# Application of the Key Characteristics of Carcinogens to Per and Polyfluoroalkyl Substances

**DOI:** 10.3390/ijerph17051668

**Published:** 2020-03-04

**Authors:** Alexis M. Temkin, Barbara A. Hocevar, David Q. Andrews, Olga V. Naidenko, Lisa M. Kamendulis

**Affiliations:** 1Environmental Working Group, Washington, DC 20009, USA; dandrews@ewg.org (D.Q.A.); olga@ewg.org (O.V.N.); 2Department of Environmental and Occupational Health, School of Public Health, Indiana University, Bloomington, IN 47405, USA; bhocevar@indiana.edu (B.A.H.); lkamendu@indiana.edu (L.M.K.)

**Keywords:** key characteristics of carcinogens, PFAS, oxidative stress, immunotoxicity, environmental carcinogen

## Abstract

Per- and polyfluoroalkyl substances (PFAS) constitute a large class of environmentally persistent chemicals used in industrial and consumer products. Human exposure to PFAS is extensive, and PFAS contamination has been reported in drinking water and food supplies as well as in the serum of nearly all people. The most well-studied member of the PFAS class, perfluorooctanoic acid (PFOA), induces tumors in animal bioassays and has been associated with elevated risk of cancer in human populations. GenX, one of the PFOA replacement chemicals, induces tumors in animal bioassays as well. Using the Key Characteristics of Carcinogens framework for cancer hazard identification, we considered the existing epidemiological, toxicological and mechanistic data for 26 different PFAS. We found strong evidence that multiple PFAS induce oxidative stress, are immunosuppressive, and modulate receptor-mediated effects. We also found suggestive evidence indicating that some PFAS can induce epigenetic alterations and influence cell proliferation. Experimental data indicate that PFAS are not genotoxic and generally do not undergo metabolic activation. Data are currently insufficient to assess whether any PFAS promote chronic inflammation, cellular immortalization or alter DNA repair. While more research is needed to address data gaps, evidence exists that several PFAS exhibit one or more of the key characteristics of carcinogens.

## 1. Introduction

Per- and polyfluoroalkyl substances (PFAS) are human-made fluorinated chemicals that have become pervasive contaminants in people and the environment. Since the 1950s, thousands of different PFAS have been produced for use in industrial and consumer products. The use of PFAS in firefighting foams and presence in industrial discharges has resulted in extensive contamination worldwide. Currently, PFAS are detectable in nearly all humans, with exposures beginning during fetal development. As detailed in this article, a growing body of research suggests that exposure to PFAS at contaminated sites and in the general population can adversely impact human health. One well-studied PFAS, perfluorooctanoic acid (PFOA) was reviewed by the International Agency for Research on Cancer (IARC) in 2016 and classified as group 2B, a possible human carcinogen [1]. Additionally, the U.S. Environmental Protection Agency (EPA) classified PFOA as well as two other PFAS, perfluoroctosulfonate (PFOS) and GenX, as having suggestive evidence of carcinogenic potential [2,3,4]. 

Recently, the Key Characteristics of Carcinogens framework was developed in order to facilitate the organization and characterization of mechanistic data for cancer hazard identification [5,6]. Using evidence from established carcinogens, ten characteristics commonly exhibited by and shared among carcinogenic agents were identified (Table 1). Importantly, these characteristics aid in cancer hazard classification through integrative evaluation with human and animal evidence of carcinogenicity. This framework is sufficiently robust to allow for the inclusion of evidence from molecular epidemiology, animal toxicity and high-throughput assay screening studies. For two of 34 agents evaluated in earlier studies, strong evidence for several key characteristics of carcinogens strengthened the overall assessment leading to their classification as a “probable human carcinogen” [5]. Most recently, in an assessment of 13 haloacetic acids, inclusion of data for the key characteristics of carcinogens in combination with other read-across information led to the recommendation of listing two chemicals as probable human carcinogens despite a lack of animal carcinogenicity studies [7].

Incorporation of a key characteristic analysis into risk assessments is increasingly utilized by U.S. and international government agencies including IARC [8], the Integrative Risk Information System at the U.S. EPA [9], the National Toxicology Program [7] and California’s Office of Environmental Health Hazard Assessment [10]. “Some perfluorinated chemicals” were also listed as a priority for IARC to review in the coming years [11]. PFAS present a unique opportunity to apply the key characteristics of carcinogens framework to individual, well studied PFAS such as PFOA and groups of structurally similar PFAS. Because of the number of PFAS compounds and the breadth of epidemiological, toxicological and mechanistic data published in the last several years, such a review is timely, practical and urgent.

This state-of-the-science review summarizes the available data for several PFAS for each of the key characteristics of carcinogens and evaluates the strength of evidence for each characteristic. We adapt the framework to include the findings from epidemiological studies. This article addresses PFOA as the most studied member of the PFAS group; long-chain PFAS that have a carbon chain length of eight or more for perfluoroalkyl carboxylic acids and six or more for perfluoroalkane sulfonates; and short-chain PFAS for which data are available for at least one of the key characteristics of carcinogens. In addition, data gaps are identified where additional research is needed. 

## 2. Existing Research on PFAS Exposure and Cancer Risk

Human biomonitoring for PFAS has been conducted by U.S. Centers for Disease Control and Prevention (CDC), by the American Red Cross [12], the state of California biomonitoring program [13], the state of New Jersey [14] and by academic laboratories and institutions in the U.S. and globally [15,16]. Initially, biomonitoring focused on two PFAS with an 8 carbon fluorinated chain, PFOA and perfluorooctanesulfonate (PFOS). Following the end of PFOS production in 2002 and the phase-out of PFOA use in the United States that started in 2006, mean serum levels of PFOA and PFOS dropped in the U.S. population from 5.21 to 1.56 ng/mL for PFOA and 30.4 to 4.72 ng/mL for PFOS from 2000 to 2016 [17]. Recent CDC biomonitoring studies also detected PFHxA, PFBA, GenX, and PFHpA in 22.6, 13.3, 1.2 and 1.1 percent of the U.S. population, respectively [18]. 

For the general population, the majority of PFAS intake has been attributed to food. Drinking water, indoor air and dust, and consumer products provide additional sources of PFAS exposure [19]. Consumption of microwaved popcorn and other preheated packaged foods, use of floss, and the presence of stain-resistant carpet or furniture coatings in the home also contribute to PFAS exposure [15,20]. In communities near facilities using or manufacturing PFAS, including military uses of PFAS-based firefighting foam, high serum PFAS levels in people are associated with PFAS contamination of water supplies [21]. For example, in Little Hocking, Ohio and the highest exposure areas of the Veneto region in Italy, median serum PFOA concentrations were 75 and 45 times higher in local residents, respectively, compared to the general population [22,23].

PFAS interactions with biological tissues and organisms are distinct from those of other environmentally persistent halogenated synthetic chemicals. Unlike organochlorine aromatic compounds such as polychlorinated biphenyls, chlorinated dioxins and dichlorodiphenyltrichloroethane (DDT), PFAS are not lipophilic and do not accumulate in adipose tissue [24]. In vivo, PFAS bind to proteins in the blood such as albumin and have been detected in tissues including the liver, pancreas, kidney, lung and brain [25,26]. PFAS readily cross the placenta and are detected in cord blood and fetal tissues, as well as in breast milk [27,28]. 

While hundreds of research studies have been published on the health harms due to individual PFAS, significant data gaps remain, particularly with respect to PFAS other than PFOA and PFOS, and for PFAS mixtures. For example, in Wilmington, North Carolina where PFAS contaminated water supplies, over 30 unique PFAS have been identified in the watershed [29], yet limited or no toxicological information is available concerning the potential health effects of nearly all of these chemicals. 

Human epidemiologic studies have linked occupational, community, and general population exposure to PFOS and PFOA with an elevated risk of cancer. Prior reviews have focused on all cancers [19,30], and on breast [31] and kidney cancer [32]. A study of occupationally exposed workers reported a 3.3-fold increase in prostate cancer mortality for each month spent in the chemical division of PFOA production [33], although a re-analysis of this cohort did not detect an association between exposure and cancer mortality or incidence [34]. In another occupational study, a positive trend based on exposure-response was observed for pancreatic and prostate cancer, although these did not reach statistical significance [35]. Most recently, an occupational cohort study in Italy found a nearly seven-fold increase in liver cancer mortality and a five-fold increase in lymphatic and hematopoietic cancer mortality in the highest tertile of PFOA exposure relative to non-chemical factory workers [36]. 

In the C8 Health Project, comprised of individuals who lived or worked in the mid-Ohio Valley where drinking water was contaminated with PFOA, elevated PFOA levels in the body were associated with greater risk of non-Hodgkin lymphoma and kidney, testicular, prostate, and ovarian cancers [37]. In a general population study, Eriksen and colleagues reported a 30%–40% increase in incidence rate ratios for prostate cancer with elevated PFOS exposure, and a positive association between plasma concentrations of PFOA and risk for pancreatic cancer [38]. In the Danish National Birth Cohort study, significantly elevated incidence of breast cancer was reported for women in the highest quintile of perfluorooctane sulfonamide (PFOSA) body burden levels compared to lowest quintile of PFOSA levels [39]. In addition, significantly higher blood levels of PFOS and PFOA were reported in 31 breast cancer cases compared to controls in Inuit women in Greenland [40]. 

Limitations in these epidemiologic studies include differences in reporting disease (incidence versus mortality) and different approaches for estimating PFAS exposure (job matrix estimates and water measurements versus direct biological serum measurements). In addition, small numbers of rare cancer types, such as pancreatic cancer, in individual cohorts limit the statistical power of the published analyses. Nevertheless, several studies have reported a significant association of PFOA and/or PFOS exposure with kidney and prostate or testicular cancer [37,38]. Due to the latency of cancer development, future follow up studies of these populations, especially community-exposed populations, may reveal additional associations with individual cancer types. 

Two-year carcinogenicity bioassays in rats have been conducted for PFOA [41,42] PFOS [43], the six carbon perfluorocarboxylic acid PFHxA [44] and HFPO-DA, also known as GenX (full abbreviation list available in Appendix A) [45]. The “tumor triad” of hepatic, Leydig cell and pancreatic acinar cell tumors was observed for PFOA and GenX, while only hepatic lesions were observed in PFOS-exposed rats and no evidence of tumorigenicity was reported PFHxA. A recently published carcinogenicity study of PFOA in rats [41] found increased incidence of pancreatic tumors in males at all doses tested (approximately 1.1 to 4.6 mg/kg/day) which were lower and administered at an earlier age than in previous studies, highlighting the importance of developmental exposure with regards to carcinogenic classification and potential. Additionally, for other toxicological endpoints, short-chain PFAS with faster elimination rates were found to be equally or more toxic than PFOA when accounting for difference in toxicokinetics [46], a finding that may be relevant for cancer risk assessment. 

## 3. Literature Search for PFAS and Key Characteristics of Carcinogens

Here, we assemble, organize and evaluate the peer-reviewed literature on PFAS through the lens of the key characteristics of carcinogens. Previous systematic reviews for toxicity data have been performed for several PFAS, such as the Agency for Toxic Substances and Disease Registry (ATSDR) toxicological profiles [47] and EPA assessments of PFOA [2], PFOS [3], PFBS [48] and GenX [4]. Here, we reference studies and conclusions from those assessments and review information from studies published more recently after the reviews were completed. Using the PubMed database, we searched the peer-reviewed literature for epidemiological, toxicological and mechanistic studies guided by search terms outlined in Guyton et al. (2018) [5] for each key characteristic in combination with “PFAS”, “perfluoro*” or names, abbreviations and Chemical Abstract Services (CAS) numbers of individual chemicals. Additional details on study inclusion and study design are given within each section. For most key characteristics we focused on mammalian studies. In circumstances where the dataset was very limited, evidence from non-mammalian species was also considered. Based on the literature search we identified and included information regarding key characteristics features of 26 different PFAS (Appendix A). Throughout the article, we refer to individual PFAS compounds by abbreviated names; full chemical names are listed in the Appendix A. 

## 4. Strength of Evidence Assessment 

For a given key characteristic, we evaluated whether the reviewed literature indicated strong evidence, suggestive evidence, no association or insufficient data for PFOA, other long-chain PFAS, and short-chain PFAS. *Strong evidence* was designated if at least two types of evidence (i.e., in vivo and in vitro) had consistent findings for multiple representative PFAS within a group, with more data-limited group members having positive evidence from one study type. If only one type of evidence provided positive findings, or if findings for distinct PFAS were inconsistent, then the overall evidence was considered *Suggestive.* If two types of evidence had studies limited in number and/or design or reported largely inconsistent findings, the data were considered *Insufficient* to make an assessment. For epidemiological literature, we utilized a qualitative assessment of evidence based on the presence or lack of a significant finding. 

## 5. Analysis of Individual Key Characteristics

### 5.1. Is Electrophilic or Can Be Metabolically Activated

A unifying characteristic of PFAS is their resistance to biodegradation, resulting from the chemical stability of the carbon-fluorine bond. This stability renders the PFAS family of chemicals non electrophilic, relatively metabolically inert and resistant to metabolic activation [47]. Most PFAS accumulate and persist in biological tissues and are excreted as unchanged compounds. The half-lives of individual PFAS in humans depend both on the carbon chain length and functional groups. For PFOA, PFOS and PFHxS, half-lives in humans range between 2.7 to 5.3 years [49,50]. Half-lives for shorter chain compounds, such as PFBS, are approximately one month [51]. In animal studies, the PFOA replacement chemical GenX does not appear to bioaccumulate and is excreted in its original structure [52]. For some PFAS, such as the 4:2, 6:2, 8:2 and 10:2 fluorotelomer alcohols (FTOHs) and polyfluoroalkyl phosphate diesters (diPAPs), these substances can be detected in the original form as well as undergo biotransformation to perfluoroalkyl acids of the same chain length and other metabolites [53,54,55,56,57]. Overall, there is strong evidence supporting that PFOA as well as long-chain and short-chain carboxylates and sulfonates are not electrophilic or metabolically activated, while FTOHs and PAPs can be metabolized into but not limited to the above-mentioned substances (Table 2).

### 5.2. Is Genotoxic

Genotoxicity is a well-studied mechanism of environmental carcinogenesis. PFAS, primarily PFOA and PFOS, have been evaluated and reviewed for genotoxicity. Prior reviews reported that PFAS are not directly mutagenic and suggested that the DNA damage observed in some assays is likely a secondary event resulting from oxidative damage [47]. Recent toxicological studies reported similar findings for legacy and long-chain PFAS as well as the replacement and short-chain PFAS, GenX and PFBS [4,48,59,60]. Additionally, 28 day toxicity studies published by the National Toxicology Program for the carboxylates, PFOA, PFHxA, PFNA and PFDA and the sulfonates, PFOS, PFHxS, and PFBS, found no indication of genotoxicity in vivo or in vitro [61,62]. Given the lack of direct genotoxicity for PFAS chemicals, any carcinogenic hazard is likely due to mechanisms other than direct DNA damage. 

### 5.3. Alters DNA Repair or Causes Genomic Instability

In addition to direct genotoxicity, xenobiotics can contribute to changes in DNA through alteration of DNA replication or repair [6]. These alterations can be mediated by modulation of specific DNA enzymes and impact on genome-wide processes such as nucleotide excision repair [63]. Our literature search did not identify studies that evaluated these endpoints for PFAS, precluding an evaluation of whether PFAS impact this key characteristic. However, epigenetic modifications and generation of reactive oxidation species, described in Section 5.4 and Section 5.5 of this article, are known to contribute to genomic instability. 

### 5.4. Induces Epigenetic Alterations

Changes in DNA methylation is a form of epigenetic regulation of gene expression implicated in the development of disease [64]. Site-specific hypermethylation in the promoter regions of tumor suppressor genes silences their expression [65,66]. Global hypomethylation throughout the genome is a characteristic of several cancers and has been associated with a poor prognosis in patients [67]. Additionally, hypomethylation affects mostly repeated sequences in the genome and can contribute to carcinogenesis by triggering transcription of oncogenes and inducing genetic instability [68]. Data concerning epigenetic effects of chemicals are becoming increasingly important for carcinogen hazard identification and are expected to become even more important in future cancer risk assessments [69]. 

PFAS, primarily PFOA and PFOS, can induce both hypo- and hypermethylation of specific gene regions and the genome globally. Guerrero-Preston et al. (2010) [70] analyzed global DNA methylation in human umbilical cord serum from 30 newborns and reported an association between PFOA levels and global DNA hypomethylation. PFOS, but not PFOA, PFNA or PFUnA was associated with global hypomethylation, measured by Alu methylation status in 363 cord blood samples from Taiwan [71]. In a Japanese cohort, prenatal PFAS exposure was not associated with cord blood global methylation status, but hypomethylation of the IGF2 promoter region was significantly associated with PFOA levels and reduced ponderal index [72]. Miura and colleagues identified several differentially methylated regions associated with prenatal PFOA and PFOS exposure in this cohort [73].

Similarly, Kingsley and colleagues examined a cohort of 44 mother–child pairs and reported no correlation between PFOA exposure during pregnancy and global DNA methylation in offspring cord blood but identified 20 PFOA specific differentially methylated CpG sites in genes involved in cell growth [74]. In males in a Faroese birth cohort, PFOS exposure was associated with changes in thousands of differentially methylated regions, especially for genes involved in embryonic growth and development [75].

In a cohort of 652 highly exposed adults from the C8 Health Project, serum PFOS concentrations, but not PFOA, PFNA or PFHxS, were associated with greater LINE-1 DNA methylation in peripheral blood leukocytes [75]. A cross-sectional analysis of adults found no association between exposure to PFOA, PFOS, PFNA, and PFHxS and global DNA methylation (LINE-1, Alu sequences) of sperm DNA from 262 men from Greenland, Poland and Ukraine [76].

A limited number of studies investigated DNA methylation changes associated with PFAS exposure in vitro while evidence in animal studies is sparse. Mice exposed to PFOS during gestation exhibited decreased global DNA methylation in the liver as well as increased methylation of the glutathione-S-transferase Pi (GSTp) promoter [77]. Similarly, in a human liver cell line, PFOA caused hypermethylation of the promoter region of GSTp and upregulation of DNA methyltransferase 3, although no changes in global DNA methylation or promoter regions of other genes were observed [78]. Global DNA methylation decreased during PFOS-induced fat cell differentiation in cells which was accompanied by an upregulation of DNA methyltransferases and demethylation of the PPARγ promoter [79]. One PFOS study showed no effects on fat cell differentiation or methylation [80], while another observed increased hypomethylation at several sites although there was no effect on cell differentiation [81]. 

Changes in DNA methylation following PFAS exposure represent one form of epigenetic regulation. Other epigenetic changes such as histone modifications, nucleosome positioning and expression of non-coding microRNAs also occur in cancer [66], and additional research is needed to evaluate the potential effects of PFAS on these types of epigenetic regulation. Data gaps remain around possible epigenetic impacts of the numerous members of the PFAS family, particularly newly introduced PFAS. Nevertheless, existing data suggest that both PFOA and PFOS may elicit differential methylation patterns similar to those associated with cancer and exposure to known carcinogens [69,81]. Overall, there is suggestive evidence for an association between PFOA and PFOS and the induction of epigenetic alterations, although there is insufficient evidence for this key characteristic for other PFAS (Table 3).

### 5.5. Induces Oxidative Stress 

Oxidative stress is defined as an imbalance between the production of reactive oxygen species (ROS) and their elimination by cellular defense mechanisms. ROS can cause damage to lipids, proteins, and DNA, and contribute to the pathology observed in chronic inflammatory conditions, such as aging and cancer [83]. Oxidative stress is a common consequence of exposure to environmental pollutants, and is a characteristic exhibited by many recognized carcinogens.

Metabolomics has been used to study human responses to internal and external stressors such as disease, and environmental exposures [84]. In a study of 181 Chinese male adults, the serum concentrations of eleven PFAS, PFOA, PFOS, PFBA, PFBS, PFDA, PFHpA, PFHxA, PFHxS, PFNA, PFUnA and PFDoA, and metabolome changes were evaluated [85]. Six PFAS, (PFOA, PFOS, PFDA, PFNA, PFUnA and PFHxS), were detected in the majority of study participants and were highly correlated with one another. The sum of all detected PFAS was used to determine the association between PFAS exposure and metabolome changes. In this study, pathways involved in oxidative/nitrosative stress, including modulation of glutathione, α-tocopherol, and ascorbate pathways, were associated with PFAS exposure [85]. As reviewed in Section 5.6, exposure to PFAS has been associated with chronic inflammation, a condition in which oxidative stress has been observed. Thus, alterations of biomarkers of oxidative stress are likely to occur in human populations with conditions of inflammation. However, no epidemiological studies have yet evaluated potential associations between PFAS exposure and biomarkers of oxidative stress.

A significant body of data from both in vivo and in vitro studies, including mammalian and wildlife species, have confirmed that PFAS are capable of inducing ROS and/or a condition of oxidative stress. Oxidative stress has been most extensively studied for PFOA and PFOS. In the HepG2 human hepatoma cell line, oxidative DNA damage (8-hydroxydeoxyguanosine; 8-OHdG) and ROS levels increased following exposure to PFOA and PFOS at 400 nM and higher [86,87]. However, PFBS and PFHxA did not increase ROS in this cell line [86]. Urinary 8-OHdG and lipid peroxidation (malondialdehyde; MDA) were also elevated in PFOA exposed rats [88]. In a separate investigation, PFOA increased ROS in HepG2 cells, and also activated caspase-9 and apoptosis [89]. PFOA induced an inflammatory response in human mast cells, as evidenced by induction of the pro-inflammatory cytokines TNFα, IL-1β, IL-6 and IL-8, and was associated with increased ROS production [90]. In mouse testis, the lowest dose tested of 2.5 mg/kg/day PFOA increased MDA and hydrogen peroxide levels, decreased the expression of the oxidant-sensitive transcription factor, Nrf2, and inhibited the activities of antioxidant enzymes (superoxide dismutase and catalase) [91]. PFOA also induces oxidative stress in a dose-dependent manner from 0.5 to 5 mg/kg/day in the mouse liver and pancreas [25], and a variety of tissues including rat pancreatic β-cells [92], mouse testes and epididymis [93], and mouse ovary [94]. Similarly, in vivo and in vitro studies have demonstrated that PFOS exposure results in oxidative stress, evidenced by the production of ROS, changes in antioxidant enzymes, elevation of lipid peroxidation products and changes in Nrf2 [32,95,96,97]. Increased lipid peroxidation and antioxidant enzyme activity were also observed in mouse testes following PFNA exposure [98]. 

As PFAS are ubiquitous in the environment, the impact of PFAS has been evaluated in wildlife species. In a comprehensive study in male Arctic seabirds (*Rissa tridactyla*), six of 20 long-chained PFAS (linear PFOS, PFNA, PFDA, PFUnA, PFDoA, PFTrDA, PFTeDA) were detected and assessed for their ability to modulate plasma biomarkers of oxidative stress, [99]. In general, the severity of oxidative stress in plasma correlated with the carbon-chain length of PFAS [99].

A few studies have evaluated whether PFAS chain length affects their ability to induce oxidative stress. Erythrocytes have an extensive antioxidant system including nonenzymatic and enzymatic antioxidants, making them a useful model to assess systemic oxidative stress. Exposure of human erythrocytes to PFOA, PFPeA, and PFDA decreased GSH levels and increased MDA content, concomitant with alterations in the activities of superoxide dismutase, catalase, and glutathione peroxidase [100]. In this study, PFDA, a 10 carbon PFAS, caused a greater level of oxidative stress in erythrocytes. Using HepG2 cells, PFOA, PFOS, PFHxS, PFNA, PFDA, PFUnA, and PFDoA, were investigated for the potential to generate ROS and modulate total antioxidant capacity (TAC). All PFAS evaluated except PFDoA increased ROS, while PFOA, PFOS and PFDoA also lowered TAC. In contrast, TAC was increased following exposure to PFHxS, PFNA and PFUnA [101]. In differentiating neurological cells, treatment with PFOA, PFOS, PFOSA and PFBS at micromolar concentrations increased oxidative stress via increased MDA production [102]. 

While only a limited number of human epidemiological studies have examined metabolic endpoints relevant to oxidative stress, initial reports describe a positive association. The existing animal and in vitro data demonstrate that several long-chain PFAS, induce oxidative stress in multiple target tissues, as well as in vitro. Therefore, there is strong evidence for an association between PFOA and other long-chain PFAS and this key characteristic (Table 4). The data are insufficient for short-chain PFAS since only three in vitro studies with equivocal findings exist for this PFAS group (Table 4). Further research aimed at measuring specific biomarkers of oxidative stress in connection with PFAS exposures in human populations as well as for short-chain PFAS would provide additional information regarding this key characteristic of carcinogens.

### 5.6. Induces Chronic Inflammation

Chronic inflammation, recognized as a hallmark of cancer, can be both a driver of cancer development as well as a feature characterizing the tumor microenvironment [103]. The association between PFAS exposure and diseases characterized by chronic inflammation has been investigated in occupational, community and general population epidemiological studies. A previous review noted inconsistency of findings amongst studies to date [104]. Here we briefly summarize these studies and include more recent publications. A significant positive association between ulcerative colitis, an autoimmune disease exhibiting chronic gastrointestinal tract inflammation, and serum PFOA levels was found in workers and in highly exposed community members participating in the C8 Health Project [105,106]. In a cross-sectional study of the C8 Health Project, the incidence of reported osteoarthritis was positively associated with serum levels of PFOA, but inversely correlated with PFOS levels [107]. A similar association for PFOA in different subpopulations was observed in a cross-sectional study of participants in the National Health and Nurtrition Examination Survey (NHANES) [108], while no association between PFOA exposure and osteoarthritis was observed in occupationally exposed workers [109]. 

In a recent study of a large prospective case-control study within the Nurses’ Health Study II, after controlling for several confounders including diet and physical activity, higher plasma concentrations of PFOA and PFOS were associated with an elevated risk for development of Type 2 diabetes [110], a chronic metabolic disorder characterized by an inflammatory environment that increases the risk for pancreatic cancer [111]. Similarly, in another prospective study of individuals enrolled in a diabetes prevention program, baseline PFOA levels were associated with a significant increased risk of diabetes in individuals who did not undergo lifestyle interventions [112] although a third prospective study in Sweden observed non-statistically significant inverse associations between the sum of six long-chain PFAS (PFOA, PFOS, PFNA, PFDA, PFUnDA, PFHxS) in plasma and risk of Type 2 diabetes, where PFOS accounted for the majority of exposure [113]. 

Chronic inflammation can cause oxidative stress (key characteristic 5), while activation of the immune system (key characteristic 7) can result in chronic inflammation. Recent animal and in vitro studies have reported PFAS-related inflammation that can promote a disbalance of overall immune homeostasis. Long-chain PFAS exposure (PFOA, PFDA, and PFUnA) in mice cause an allergic inflammatory response [114,115]. PFOA can also alter T cell profiles and expression of proinflammatory cytokines in the spleen of mice [116], and PFOS, PFOA and PFDA induced the release of inflammatory cytokines in cells [117,118]. In human bronchial epithelial cells, proinflammatory cytokine release resulted following exposure to PFOA and PFOS but not PFBS, PFHxS and the fluorotelomer 8:2 FTOH [119]. Interestingly, inflammation may mediate PFAS transfer across the blood cerebrospinal fluid barrier in humans through enhanced barrier permeability [120].

Overall, there is somewhat suggestive evidence from epidemiological, animal and in vitro studies that PFOA, PFOS, PFDA and PFUnA may promote chronic inflammation, yet the number of studies are limited and biomarkers of chronic inflammation and identification of inflammatory diseases which are specific to PFAS exposures are needed to more fully characterize the relationship between exposure, chronic inflammation and risk for cancer development (Table 5). Data available for short-chain PFAS was limited to one in vitro study of PFBS (Table 5). Therefore, with the research studies currently available, there is insufficient evidence for all PFAS and this key characteristic.

### 5.7. Is Immunosuppressive

Immunotoxicity is a sensitive endpoint for PFAS toxicity, with implications for children’s health and for lifetime cancer risk (reviewed in [121]). Both immunosuppression and immunoenhancement have been observed in animal models and well documented in the epidemiologic literature, primarily in children based on early life and in utero exposures to PFAS. Immunosuppression can be defined as the reduced ability of the immune system to effectively respond to foreign antigens, including antigens present on tumor cells [6]. Immunosuppression can result in decreased responses to vaccinations, and more significantly, can allow transformed cancer cells to escape immunosurveillance. Immunoenhancement can be defined as inappropriate activation of the immune system that can trigger hypersensitivity reactions such as allergies or autoimmune diseases and contribute to chronic inflammation, another key characteristic of carcinogens [121].

Regulatory agencies typically rely on the T cell-dependent antibody response (TDAR) assay in laboratory animals to evaluate immunosuppression [122]. This assay, which involves injection of a foreign antigen followed by analysis of collected blood for presence of IgM antibodies, is a measure of the function of the adaptive immune system requiring both T and B cell participation to generate a response. A second antigen injection can be performed followed by analysis for IgG antibodies. In human studies, this assay can be approximated by investigating the response of exposed individuals to vaccinations. Both prospective cohort and cross-sectional epidemiologic studies have documented the association of PFAS with decreased antibody levels in response to vaccination. In children in the Faroe Islands exposed to PFAS in utero, a two-fold increase in PFOS and PFOA serum concentrations, where the means serum levels were 27.3 and 3.20 ng/mL respectively, was associated with exhibiting antibody levels below protective levels for tetanus and diphtheria vaccinations [123]. When PFHxS concentrations were included with PFOA and PFOS in the analysis of this cohort, a stronger negative association was found [124]. A prospective general population study from the Norwegian Mother and Child Cohort found that the concentration of PFOA, PFOS, PFHxS and PFNA in maternal blood at time of delivery, which were lower than concentrations measured in the Faroese cohort, correlated with decreased levels of anti-rubella antibodies in their three-year-old children [125]. In adult C8 Health Project participants, higher serum concentrations of PFOA were associated with lower antibody titers following A/H3N2 influenza vaccination [126]. In a cross-sectional study of NHANES data, increased serum levels of PFOA and PFOS were associated with decreased anti-mumps antibodies while PFOA, PFOS and PFHxS concentrations were associated with lower anti-rubella antibodies [127]. 

Allergies are classified as Type I hypersensitivity reactions that can be monitored in humans by the presence of increased serum levels of IgE. While prospective studies of birth cohorts did not reveal a consistent association of PFAS exposure and Type I hypersensitivity reactions in children or serum IgE levels [121], analysis of NHANES data identified an association of PFOA and PFOS with respiratory hypersensitivity reactions [127]. In addition, PFOA levels were associated with total serum IgE in this cohort [127]. A case-control study in Taiwan found a positive correlation with PFOA and PFOS and asthma severity scores as well as serum IgE in children [128]. An association was found between PFAS levels and asthma at age five and thirteen in children from the Faroe Islands who did not receive an MMR vaccination [129]. Autoimmune diseases are classified as Type IV hypersensitivity reactions and can be difficult to diagnose in humans. As noted above, the incidence of ulcerative colitis was increased in adults in the C8 Health Project with increased PFOA exposure [106]. Elevated serum PFOA levels in workers also show correlation with ulcerative colitis and rheumatoid arthritis [109]. 

Rodent studies have demonstrated that both PFOA and PFOS are capable of suppressing the TDAR, with lower bound benchmark dose estimates of 1.75 mg/kg/day, which supports the epidemiologic findings that these PFAS elicit immunosuppression [130,131,132]. In vivo animal studies evaluating the immunosuppressive potential of other PFAS are sparse. A recent report of oral exposure to PFDA in mice and rats observed a reduction of immune cell populations in the spleen of treated mice, although the assay did not detect overt immunosuppression [133]. However, exposure of mice to 10 mg/kg/day 8:2 fluorotelomer was found to decrease inflammatory cytokines in serum and mRNA levels in the thymus, liver and spleen and in vitro fluorotelomer treatment of primary splenocytes altered conconavalin-stimulated proliferation and IFNγ secretion [134]. In vitro studies utilizing human leukocytes found that PFOA and PFOS reduced natural killer (NK) cell activity and also reduced lipopolysaccharide (LPS)-induced release of the pro-inflammatory cytokine TNFα [135]. In vitro studies designed to investigate the mechanism of PFAS-mediated immunosuppression also found that concentrations as low as 0.1 ug/mL PFBS, PFOSA, PFDA and fluorotelomer as well as PFOA and PFOS, suppressed the release of TNFα following LPS stimulation of human peripheral blood leukocytes, which could be linked to decreased LPS-induced NF-κB activation [136,137]. 

Animal studies on PFAS-mediated immunoenhancement are limited. In a mouse model of asthma involving dermal exposure to ovalbumin (OVA) as the antigenic trigger, PFOA did not increase serum IgE levels but increased the OVA-mediated hypersensitivity response [138]. In contrast, no increase in OVA-mediated hypersensitivity was found following a single-dose exposure to PFOA or PFOS [139] while 60 days oral exposure to PFOS was shown to increase antigen-specific IgE serum levels in mice [140]. 

The National Toxicology Program evaluated the immunotoxicity potential for PFOA and PFOS and concluded that the evidence for suppression of antibody responses was high for experimental animals and moderate in humans [141]. Studies published since the completion of the NTP review reinforce the evidence of PFAS-associated immunotoxicity. Thus, despite data gaps for many PFAS, existing data from epidemiological, toxicological, and in vitro studies are consistent overall and support the finding that exposure to different long-chain PFAS (PFOA, PFOS, PFHxS, PFNA, 8:2 fluorotelomer) and the short-chain PFBS (in vitro) results in immunosuppression (Table 6). We conclude that there is strong evidence for an association for PFOA and long-chain PFAS and suggestive evidence for short-chain PFAS and this key characteristic. 

### 5.8. Modulates Receptor-Mediated Effects

Alterations in cellular receptor signaling and subsequent changes in genomic and metabolic pathways controlled by those receptors have a profound influence on the development, and function of cells, tissues and organisms as well as on the risk of cancer and other diseases [6,142,143]. For example, exposure to xenobiotic substances that modulate estrogen and testosterone signaling pathways can influence breast and testicular cancer development [144,145,146,147]. Activation or antagonism of hormone receptors, changes in circulating hormones or other hormonally relevant endpoints such as anogenital distance (AGD) can be used to evaluate receptor-mediated effects of potential carcinogens [6,148]. Additionally, activation of the peroxisome proliferator active receptor alpha (PPARα) is implicated in rodent hepatocarcinogenesis, with possible relevance for human cancer risk [149].

Here we review epidemiological, toxicological and mechanistic data that help elucidate how PFAS substances modulate receptor-mediated effects, with a primary focus on recent studies and early life exposures because of children’s greater susceptibility to the effects of endocrine disruptors [150], and children’s higher exposure to PFAS than adults [151]. New data published since an earlier review of mainly cross-sectional studies in adults [152] shed additional light on the topic of PFAS and receptor-mediated effects associated with reproductive outcomes.

In a Danish cohort of men exposed to PFAS in utero, maternal PFOA serum levels (median = 3.8 ng/mL), but not PFOS was associated with reduced sperm concentration and count, and increased levels of luteinizing hormone and follicle stimulating hormone [153]. In a Japanese cohort, in utero PFOA and PFOS exposure were positively correlated with umbilical cord serum estradiol and negatively correlated with testosterone/estradiol ratio, and progesterone in males [154]. Reduced circulating levels of testosterone and estradiol were observed in children age 6 to 9 exposed to PFOA and PFOS and reduced levels of insulin-like growth factor-1 were observed in children exposed to PFNA and PFOS [155]. Reduced serum testosterone also correlated with PFOA, PFOS, and PFUnA, exposure in two Taiwanese cross-sectional adolescent studies [156,157]. PFOA, PFOS, PFDA and PFNA were also associated with increased estradiol levels among children with asthma [157].

A study of 18–22-year-old men in the PFAS-contaminated Veneto region of Italy, observed reduced testicular volume in males and increased circulating levels of luteinizing hormone and testosterone, and shorter AGD compared to controls [158]. In two mother–child cohorts from Denmark and Shanghai, shorter AGD was associated with exposure to PFOS, PFHxS, PFNA and PFDA in females at 3 months of age [159] and PFOS, PFDA, and PFUnA in males at birth; at six months of age, this association was observed only for PFOS [160].

PFAS exposure also affected birth outcomes controlled by thyroid hormone [161] and increased thyroid disease at general population exposure levels [162]. A recent systematic review identified evidence of a positive association between maternal levels of PFHxS, PFOS, and PFNA and maternal thyroid stimulating hormone (TSH) and a potential negative association with T4 [163]. Additionally, PFNA exposure in males was positively associated with elevated TSH levels [163]. 

Similar to human studies, animal studies consistently documented impact on receptor-mediated pathways such as changes in circulating hormone levels and potential anti-androgen effects. Toxicological studies of PFOA in laboratory rats observed decreases in serum testosterone and increases in estradiol [164]. Similarly, PFOA-treated male mice exhibited dose-dependent reductions in testosterone and progesterone levels, reduced sperm quality, damaged seminiferous tubules and increased testicular oxidative stress at doses of 0.31 to 20 mg/kg/day [91,165]. Consistent findings were observed following in utero exposure to PFOA [166]. Male reproductive toxicity, including effects on hormone levels and fertility was observed in PFNA treated mice [98] and gestational exposure to GenX and PFHxS resulted in lower testis weight and a weak but significant association with nipple retention in males, respectively [167,168]. 

In female mice, PFOA delayed mammary gland development in dams as well as female offspring [169,170] at concentrations as low as 0.01 mg/kg/day. Research currently in progress has suggested that GenX may also impact mammary gland development in mice [171]. PFAS-induced changes in mammary glands may have important implications for breast cancer development although research on human exposure to PFAS and the risk of breast cancer demonstrated mainly positive, but some conflicting results [31,38,39,40,172]. In an early life stage mouse uterotrophic assay, low-dose exposure to PFOA (0.01 mg/kg/day) increased uterine weight concomitant with histopathological changes in female reproductive organs [173]; no uterotrophic response to PFOA was observed in adult mice [174]. In rats, PFOA and PFOS increased serum estradiol level and protein expression of estrogen receptor alpha in uterus [175].

Analysis of PFOA activity in high-throughput screening assays from the ToxCast and Tox21 programs identified that PFOA induced activity in four different estrogen receptor assays, seven peroxisome proliferator activated receptors (PPAR) and retinoid X receptor assays, one androgen receptor assay, as well as several metabolic enzyme assays [176]. PFNA, PFOS and PFHxS also activated PPARγ and estrogen receptor alpha in ToxCast assays and modulated expression of estrogen receptor-related transcriptional profiles in livers from mice exposed to these PFAS [177]. PFOA interacts with the ligand binding domain of estrogen receptors and while GenX had weaker estrogen binding ability, similar hexafluoropropylene homologues exhibited stronger affinity [178]. PFAS of varying chain lengths and structure, from C4 to C12 carboxylates and C4, C6 and C8 sulfonates were shown to intracellularly activate human and mouse PPARα [179,180]. Other studies indicate that PFOA may exert anti-androgenic effects through reduced testosterone binding to androgen receptor [158]. 

In estrogen receptor alpha binding assays, eight PFAS—PFOA, PFOS, PFHxS, PFHxA, PFBS, PFBA, 6:2 FTOH and 4:2 FTOH—had a higher affinity for the human receptor compared to the rat receptor which was increased in longer chain compounds relative to shorter chains with IC50s ranging from 16.72 to 74.27 µM [175]. Estrogenic activity was observed for PFOA, PFOS and PFHxS and antiandrogenic activity for PFHxS, PFOS, PFOA, PFNA and PFDA alone and in a mixture [181]. In another study, PFOA, PFOS, PFHxS, PFHxA, PFBS, PFBA, PMOH/GenX and PMPP/ADONA did not activate estrogen or androgen receptors at non-cytotoxic concentrations [182] and GenX, did not agonize or antagonize estrogen, androgen or glucocorticoid receptors in vitro [167]. PFOS exposure resulted in increased estradiol and decreased testosterone in in vitro steroidogenesis assays [183].

Notably, two studies examined several PFAS chemicals used in food packing materials including fluorotelomer alcohols, 4:2 FTOH, 6:2 FTOH, 8:2 FTOH, and polyfluorinated alkyl phosphate esters, 8:2 monoPAP, 8:2 diPAP, 8:2 triPAP, 10:2 diPAP and reported increased estradiol, decreased testosterone production and estrogen receptor activation but no effect on PPARα or androgen receptor [184,185]. 

PFOA and other PFAS may modulate hormonal signaling and endogenous circulating hormone levels through interaction with serum proteins that facilitate the transport of various endogenous and exogenous substances, including hormones and xenobiotic compounds [186,187]. For example, most PFAS, but not fluorotelomers, bind thyroid hormone transport proteins such as transthyretin [188,189]. Exposure to PFOA and PFOS reduced T4 levels in adult rats as well as circulating T3 and T4 levels in pregnant dams and/or offspring exposed to PFOS, PFHxS, PFBS, and GenX [167,168,190,191,192]. PFOA and PFHxA exposure also induced similar changes in T4, T3 and TSH in male rats [61]. For PFOA and PFOS, reduced circulating levels of T4 may be due changes in uptake of T4 as demonstrated in rat hepatocytes [193].

Overall, the epidemiologic evidence suggests that long-chain PFAS exposure (PFOS, PFOS, PFNA, PFHxS, PFNA, PFDA) can modulate circulating levels of several hormones. In animal bioassays, PFAS-associated male reproductive toxicity and impact on thyroid hormones are consistently reported. Finally, mechanistic data indicate that several PFAS activate PPARα, weakly activate estrogen receptor, antagonize androgen receptor and bind thyroid hormone transport proteins. Of the 26 PFAS reviewed here, 22 exhibited receptor mediated effects in vitro for at least one receptor and nine, including both long-chain and short-chain PFAS have support from multiple lines of evidence (Table 7). Therefore, strong evidence exists for the ability of PFAS to elicit receptor mediated effects. 

### 5.9. Causes Immortalization 

Immortalization of cancerous cells through inhibition of normal cellular senescence and unlimited replicative potential results in uncontrolled proliferation and facilitates both the formation of macroscopic tumors and tumor metastasis [194]. As noted in the Key Characteristics of Carcinogens framework by Smith et al. [6], immortalized cells are not subject to the Hayflick limit for the number of successive cell growth-and-division cycles, which is controlled in senescent, non-immortalized cells due to DNA damage or shortened telomeres resulting in cell death. 

Both telomere length and mitochondrial DNA content are sensitive to environmental stressors and contaminants [195,196]. To date, only a few studies have examined the influence of PFAS on endpoints related to DNA stability and immortalization. Leukocyte telomere length was inversely associated with elevated PFOS and PFDA levels in female but not male newborns [197] and with serum PFOA levels in adults 50–65 years old [198]. In the same adult study, serum PFOS and PFHxS levels were positively and negatively associated with mitochondrial DNA content respectively [198]. Intriguingly, a study of PFAS in wild Arctic seabirds found a positive relationship between PFAS and elongated telomere length in birds bearing the highest concentrations of six PFAS; PFOS (linear), PFNA, PFDA, PFUnA, PFDoA and PFTrDA [199]. 

The data are insufficient to determine how PFAS may impact immortalization due to the limited number of studies available. Further research is needed to provide information on the mechanisms by which PFAS impact telomeres and how those changes may contribute to cellular immortalization and cancer development. It is possible that other key characteristics such as induction of oxidative stress and chronic inflammation may contribute to this type of cellular transformation.

### 5.10. Alters Cell Proliferation, Cell Death or Nutrient Supply

Lack of normal cell cycle control, uncontrolled proliferation and ultimately metastasis represent a hallmark of cancer promotion and progression, which, in turn, has stimulated interest in targeting cell cycle regulatory proteins for cancer therapeutics [200]. Additionally, exogenous chemicals may contribute to tumor progression, particularly in hormone sensitive cancers [201].

In vitro, the ability of PFAS to influence cell proliferation, particularly in assays of breast and other endocrine mediated cells and tissues has been assessed. PFOA and PFOS both increased proliferation, cell migration and invasion of normal human breast epithelial cells and altered the expression of proteins involved in cell cycle regulation, including cyclin dependent kinase 4 [202,203]. Similarly, in a mixture with two other chemicals found in consumer products, environmentally relevant concentrations of PFOA increased cell proliferation in non-malignant breast cancer cells through cell cycle disruption, reduced cell apoptosis and increased estrogen receptor alpha levels [204].

In vitro, PFOA induced cell migration and invasion in human endometrial [205] and ovarian cancer cells [206], mediated by the ERK (extracellular signal-regulated kinase)/mTOR (mammalian target of rapamycin) and the ERK 1/2 nuclear factor-kappaB signaling pathways respectively. In the granulosa tumor spheroid culture model of ovarian cancer, PFOA and PFOS both increased cell proliferation mediated by insulin-like growth-factor 1 receptor [207]. In a human liver cell line, PFOA and PFOS increased the expression of cyclins and cyclin-dependent kinases and induced cell proliferation through G1 to S phase transitions [208,209]. Additionally, 6:2 and 8:2 FTOH stimulated proliferation in the MCF-7 breast cancer cell line [210]. 

Information for this key characteristic can be gathered from pathological assessments of tissues from exposed laboratory animals using tissue hyperplasia is an indicator of cell proliferation. After 28 day exposure in rats, PFOA induced hyperplasia of the olfactory and respiratory epithelium. Impact on olfactory tissue were also observed for PFHxA, PFBS and PFHxS. In addition, PFBS and PFNA induced hyperplasia in the forestomach epithelium [61,62]. 

Increased cell proliferation has been observed in vitro for PFOA, PFOS, 6:2 and 8:2 FTOH and in vivo for PFOA, PFOS, PFHxA, PFBS, PFHxS and PFNA providing suggestive evidence that both long and short-chain PFAS are associated with this key characteristic (Table 8). As yet, information on the effects of other members of the PFAS family on proliferation, nutrient supply and cell death is not available. Given that such assays can be carried out with a high-throughput in vitro approach, we anticipate that research over the next several years will fill crucial data gaps in this area.

## 6. Discussion 

Here, we present a novel review of PFAS utilizing the Key Characteristics of Carcinogens framework. Our state-of-the-science review incorporates data from epidemiological studies with supporting information from animal toxicological studies and in vitro mechanistic studies and includes data from 26 different PFAS compounds including long- and short-chain perfluoroalkyl carboxylates and sulfonates, fluorotelomer alcohols, polyfluoroalkyl phosphate esters and fluoropolyether carboxylates (Table 2, Table 3, Table 4, Table 5, Table 6, Table 7 and Table 8; Appendix A). Overall, we identified strong evidence indicating that PFOA, several long-chain PFAS and short-chain PFAS can modulate receptor-mediated effects. Similarly, strong evidence exists for the induction of oxidative stress and suppression of the immune response due to PFOA and several long-chain PFAS. For the short-chain PFAS, suggestive evidence exists for PFBS and immunosuppressive effects while data from in vitro studies measuring oxidative stress were equivocal for PFBS, PFHxA and PFPeA. We also found suggestive evidence that some PFAS, PFOA and PFOS can induce epigenetic alterations and that PFOA and PFOS as well as the long-chain PFAS, PFHxS and PFNA, and the short-chain PFAS, PFBS, PFHxA, and 6:2 FTOH, may influence cell proliferation pathways. There was insufficient evidence to assess whether PFAS can promote chronic inflammation, cause immortalization or alter DNA repair, and evidence that PFAS are not genotoxic and generally do not undergo metabolic activation with exception of certain functional groups. 

As this is a hazard-based approach, exposure dose was not one of the key factors in the strength of evidence assessment. Future research on specific key characteristics and effects at environmentally relevant doses (i.e., epidemiological studies and low dose animal studies) will be particularly important for risk assessment. More research is also needed to address the mechanistic links between the KCs and the combination of KCs and their relationship to the carcinogenic potency of chemicals that exhibit specific key characteristics. 

Additionally, differences in evidence strength and conclusions for the groups of PFAS evaluated here may be due to limited data availability rather than toxicological differences. The strength of evidence assessments for PFAS are based on the published, peer-reviewed studies. The number of such studies is significant for PFOA and PFOS, moderate for other long-chain PFAS, and limited for short-chain PFAS and certain structural functional groups such as fluorotelemers (FTOHs) and flouralkyl phosphate esters (PAPs). However, while perfluroalkyl acids are not metabolized, the PAPs and FTOHs are biotransformed in the body and environment to their corresponding carboxylates of the same chain length. Therefore, it may be appropriate to evaluate this group of PFAS by their biotransformation products. Importantly over 600 different PFAS were reported in active use over the past decade, yet for the majority of those compounds there is no toxicological information available. The present work can assist in addressing those data gaps with respect to cancer pathways that are likely to be impacted by PFAS. 

Future research into the key characteristics of carcinogens may elucidate how combinations of key characteristics influence cancer risk. In a study of 86 agents known to cause cancer in humans, on average, each agent exhibited four key characteristics [211]. This is particularly important for assessing the adverse health effects of exposure to chemical mixtures, given that mixtures of different chemicals may contribute to cancer development [212,213]. Recently, research by the Halifax Project environmental mixtures task force highlighted that chemicals commonly present in human environment cause a variety of effects associated with hallmarks of cancer [194,214]. Similar to the Key Characteristics of Carcinogens framework, the Hallmarks of Cancer framework identifies key features of tumor initiation, progression and malignancy—features that can be triggered by everyday chemical exposures [213]. As documented in this state-of-the-science review, different PFAS chemicals exhibit multiple key characteristics of carcinogens and PFAS exposure can lead to a variety of Hallmarks of Cancer phenotypes, such as inflammation, dysregulated metabolism, change in intra- and extra-cellular signaling as well as weakening or suppression of immune response which is essential for defense against cancer. Given that exposure to more than one PFAS occurs in the general population, this class of chemicals is well suited for mixtures analysis.

## 7. Conclusions

Evidence exists that multiple PFAS exhibit several of the key characteristics of carcinogens with each of 26 chemicals identified in our review exhibiting at least one characteristic, particularly receptor-mediated effects. Well-studied PFAS, including PFOA and PFAS, exhibit up to five key characteristics which can serve as research priorities for less-studied PFAS as well as sources of information for potential application of read-across approaches to unstudied endpoints for other PFAS class members. Despite the existing data gaps, both in human epidemiology and mechanistic aspects of PFAS toxicity, development of a systematic framework for understanding the PFAS class can be facilitated with the key characteristics approach and may be useful for categorizing the carcinogenic hazards of PFAS that have not undergone systematic testing.

## Figures and Tables

**Table 1 ijerph-17-01668-t001:** The ten key characteristics of carcinogens ^a^.

Key Characteristics	Examples of Relevant Evidence
1—Is electrophilic or can be metabolically activated	Parent compound or metabolite with an electrophilic structure (e.g., epoxide, quinone, etc.), formation of DNA and protein adducts
2—Is genotoxic	DNA damage (DNA strand breaks, DNA-protein cross-links, unscheduled DNA synthesis), intercalation, gene mutations, cytogenetic changes (e.g., chromosome aberrations, micronuclei)
3—Alters DNA repair or causes genomic instability	Alterations of DNA replication or repair (e.g., topoisomerase II, base-excision or double-strand break repair
4—Induces epigenetic alterations	DNA methylation, histone modification, microRNAs
5—Induces oxidative stress	Oxygen radicals, oxidative stress, oxidative damage to macromolecules (e.g., DNA, lipids)
6—Induces chronic inflammation	Elevated white blood cells, myeloperoxidase activity, altered cytokine and/or chemokine production
7—Is immunosuppressive	Decreased immunosurveillance, immune system dysfunction
8—Modulates receptor-mediated effects	Receptor in/activation (e.g., ER, PPAR, AhR) or modulation of endogenous ligands (including hormones)
9—Causes immortalization	Inhibition of senescence, cell transformation
10—Alters cell proliferation, cell death or nutrient supply	Increased proliferation, decreased apoptosis, changes in growth factors, energetics and signaling pathways related to cellular replication or cell-cycle control, angiogenesis

^a^ Reproduced from *Environmental Health Perspectives* with permission from the authors [6]. Abbreviations: ER, estrogen receptor; PPAR, peroxisome-proliferator-activated receptor; AhR, aryl hydrocarbon receptor.

**Table 2 ijerph-17-01668-t002:** Chemical specific study findings for key characteristic one—is electrophilic or can be metabolically activated.

Chemical Groups and Abbreviations	^a^ Study Findings from Lines of Evidence for KC 1—Is Electrophilic or Can Be Metabolically Activated	
	Epidemiological	Animal Bioassay
PFOA	No assocaition: reviewed in [2]; reviewed in [47]; [49]; [50]	No Association: reviewed in [2]; reviewed in [47]; [58]
**Long-chain PFAS ^b^**		
PFOS	No association: reviewed in [3]; [49]; [50]; reviewed [47]	No Association: reviewed in [3]; [58]; reviewed in [47]
PFHxS	No association: [49]; [50]; reviewed [47]	No association: reviewed in [47]
PFNA	No association: reviewed [47]	No association: reviewed in [47]
PFDA	No association: reviewed in [47]	No association: reviewed in [47]
PFUnA		No association: reviewed in [47]
PFDoA		No association: reviewed in [47]
PFOSA		No association: reviewed in [47]
8:2 FTOH		* Association: [55]
8:2 diPAP		* Association: [56]
10:2 diPAP		* Assocaition: [56]
**Short-chain PFAS ^c^**		
PFBS	No association: [51]	No association: reviewed in [47]
PFHxA		No association: reviewed in [47]
PFBA		No association: reviewed in [47]
PFHpA		No association: reviewed in [47]
GenX (HFPO-DA); PMOH		No association: [52]
6:2 FTOH	* Association: [57]	
4:2 diPAP	* Assocaition: [53]	* Association: [56]
6:2 diPAP	* Association: [53]	* Association: [56]

^a^ Data were not identified from in vitro studies. ^b^ The following long-chain PFAS are not presented in the table since no data were avaible to assess this key characteristic: PFTrDA, PFTeDA, 8: monoPAP, 8:2 TriPAP. * PFAS listed as having an “Association” with this key characteristic undergo metabolic activation but are not electrophilic. ^c^ The following short-chain PFAS are not presented in the table since no data were avaible to assess this key characteristic: PFPeA, PMPP/ADONA, 4:2 FTOH.

**Table 3 ijerph-17-01668-t003:** Chemical specific study findings for key characteristic four—induces epigenetic alterations.

Chemical Groups and Abbreviations	Study Findings from Lines of Evidence for KC 4—Induces Epigenetic Alterations		
	Epidemiological	Animal Bioassay	In Vitro
PFOA	Association: [70]; [72]; [73]; [74] No association: [71]; [82]; [76]		Association: [78]
Long-chain PFAS ^a,b^			
PFOS	Association: [71]; [73]; [82] No association: [70]; [76]	Association: [77]	Association: [79]; [81] No association: [80]
PFHxS	No association: [82]; [76]		
PFNA	No association: [71]; [82]; [76]		
PFUnA	No association: [71]		

^a^ The following long-chain PFAS are not presented in the table since no data were avaible to assess this key characteristic: PFDA, PFDoA, PFTrDA, PFTeDA, PFOSA, 8:2 FTOH, 8:2 monoPAP, 8:2 diPAP, 8:2 triPAP, 10:2 diPAP; ^b^ The following short-chain PFAS are not presented in the table since no data were avaible to assess this key characteristic: PFBS, PFHxA, PFBA, PFPeA, PFHpA, GenX (HFPO-DA); PMOH, PMPP/ADONA, 4:2 FTOH, 6:2 FTOH, 4:2 diPAP, 6:2 diPAP.

**Table 4 ijerph-17-01668-t004:** Chemical specific study findings for key characteristic five—induces oxidative stress.

Chemical Groups and Abbreviations	Study Findings from Lines of Evidence for KC 5—Induces Oxidative Stress		
	Epidemiological	Animal Bioassay	In Vitro
PFOA	Association: [85]	Association: [88]; [91]; [25]; [92]	Association: [86]; [87]; [89]; [90]; [93]; [94]; [100]; [101]; [102]
Long-chain PFAS ^a^			
PFOS	Association: [85]	Association: [95]; reviewed in [32]; reviewed in [97]; [99]	Association: [86]; [96]; reviewed in [32]; reviewed in [97]; [100]; [101]; [102]
PFHxS	Association: [85]		Association: [101]
PFNA	Association: [85]	Association: [98]; [99]	Association: [101]
PFDA	Association: [85]	Association: [99]	Association: [101]
PFUnA	Association: [85]	Association: [99]	Association: [101]
PFDoA		Association: [99]	Association: [101]
PFTrDA		Association: [99]	
PFTeDA		Association: [99]	
PFOSA			Association: [101]
Short-chain PFAS ^b^			
PFBS			Association: [102] No association: [86]
PFHxA			No association: [86]
PFPeA			Association: [100]

^a^ The following long-chain PFAS are not presented in the table since no data were avaible to assess this key characteristic: 8:2 FTOH, 8:2 monoPAP, 8:2 diPAP, 8:2 triPAP, 10:2 diPAP. ^b^ The following short-chain PFAS are not presented in the table since no data were avaible to assess this key characteristic: PFBA, PFHpA, GenX (HFPO-DA); PMOH, PMPP/ADONA, 4:2 FTOH, 6:2 FTOH, 4:2 diPAP, 6:2 diPAP.

**Table 5 ijerph-17-01668-t005:** Chemical specific study findings for key characteristic six—induces chronic inflammation.

Chemical Groups and Abbreviations	Study Findings from Lines of Evidence for KC 6—Induces Chronic Inflammation		
	Epidemiological	Animal Bioassay	In Vitro
PFOA	Association: [105]; [106]; [107]; [108]; [110]; [112] No association: [109]	Association: [115]; [116]	Association: [90]; [117]
Long-chain PFAS ^a^			
PFOS	Association: [110] No association: [107]; [113]	Association: [114]	Association: [117]; [119]
PFHxS			No association: [119]
PFDA		Association: [114]; [118]	Association: [118]
PFUnA		Association: [114]	
8:2 FTOH			No association: [119]
Short-chain PFAS ^b^			
PFBS			No association: [119]

^a^ The following long-chain PFAS are not presented in the table since no data were avaible to assess this key characteristic: PFNA, PFDoA, PFTrDA, PFTeDA, PFOSA, 8:2 monoPAP, 8:2 diPAP, 8:2 triPAP, 10:2 diPAP; ^b^ The following short-chain PFAS are not presented in the table since no data were avaible to assess this key characteristic: PFHxA, PFBA, PFPeA, PFHpA, GenX (HFPO-DA); PMOH, PMPP/ADONA, 4:2 FTOH, 6:2 FTOH, 4:2 diPAP, 6:2 diPAP.

**Table 6 ijerph-17-01668-t006:** Chemical specific study findings for key characteristic seven—is immunosuppressive.

Chemical Groups and Abbreviations	Study Findings from Lines of Evidence for KC 7—Is Immunosuppressive		
	Epidemiological	Animal Bioassay	In Vitro
PFOA	Association: [123]; [125]; [126]; [127]	Association: reviewed in [141]; [132]; [130]	Association: [135]; [136]
Long-chain PFAS ^a^			
PFOS	Association: [123]; [125]; [127]	Association: reviewed in [141]; [131]; [132]	Association: [135]; [136]
PFHxS	Association; [125]; [127]		
PFNA	Association: [125]		
PFDA		Association: [133]	Association: [137]
PFOSA			Association: [137]
8:2 FTOH		Association: [134]	Association: [134]; [137]
Short-chain PFAS ^b^			
PFBS			Association: [137]

^a^ The following long-chain PFAS are not presented in the table since no data were avaible to assess this key characteristic: PFUnA, PFDoA, PFTrDA, PFTeDA, 8:2 monoPAP, 8:2 diPAP, 8:2 triPAP, 10:2 diPAP. ^b^ The following short-chain PFAS are not presented in the table since no data were avaible to assess this key characteristic: PFHxA, PFBA, PFPeA, PFHpA, GenX (HFPO-DA); PMOH, PMPP/ADONA, 4:2 FTOH, 6:2 FTOH, 4:2 diPAP, 6:2 diPAP.

**Table 7 ijerph-17-01668-t007:** Chemical specific study findings for key characteristic eight—modulates receptor-mediated effects.

Chemical Groups and Abbreviations	Study Findings from Lines of Evidence for KC 8—Modulates Receptor-Mediated Effects		
	Epidemiological	Animal Biosassay	In Vitro
PFOA	Association: [153]; [154]; [155]; [156]; [157]	Association: reviewed in [164]; [91]; [165]; [166]; [169]; [170]; [173]; [175]; [191]; [61] No association: [174]	Association: [176]; [158]; [175]; [181]; [188]; [189]; [193]; [179]; [180] No association: [182]
Long-chain PFAS ^a^			
PFOS	Association: [154]; [155]; [156]; [157]; [160]; reviwed in [163] No Association: [153]	Association: [191]; [192]	Association: [177]; [175]; [181]; [188]; [189]; [193]; [179]; [180] No association: [182]
PFHxS	Association: [159]; reviwed in [163]	No association: [168]	Association: [177]; [175]; [181]; [188]; [189]; [179]; [180] No association: [182]
PFNA	Association: [155]; [157]; [159]; reviewed in [163]	Association: [98]	Association: [177]; [181]; [188]; [189]; [179]; [180]
PFDA	Association: [157]; [159]; [160]		Association: [181]; [188]; [189]; [179] No Association: [180]
PFUnA	Association: [156]; [160]		Association: [188]; [189]; [180] No association: [179]
PFDoA			Association: [188]; [189]; [179] No association: [180]
PFTrDA			Association: [189]
PFTeDA			Association: [188]; [189]; [179]
8:2 FTOH			Association: [184]; [185] No association: [188]
8:2 monoPAP			Association: [184]; [185]
8:2 diPAP			Association: [184]
8:2 triPAP			Association: [184]
10:2 diPAP			Association: [184]
Short-chain PFAS ^b^			
PFBS		Association: [190]	Association: [175]; [188]; [189]; [179]; [180] No association: [182]
PFHxA		Association: [61]	Association: [175]; [188]; [189]; [179]; [180] No association: [182]
PFBA			Association: [175]; [189]; [179]; [180] No association: [182]
PFPeA			Association: [179]; [180]
PFHpA			Association: [188]; [189]; [179]; [180]
GenX (HFPO-DA); PMOH		Association: [167]; [171]	Association: [178] No association: [182]
PMPP/ADONA			No association: [182]
4:2 FTOH			Association: [175]; [185]
6:2 FTOH			Association: [175]; [185] No association: [188]

^a^ The following long-chain PFAS are not presented in the table since no data were avaible to assess this key characteristic: PFOSA. ^b^ The following short-chain PFAS are not presented in the table since no data were avaible to assess this key characteristic: 4:2 diPAP, 6:2 diPAP.

**Table 8 ijerph-17-01668-t008:** Chemical specific study findings for key characteristic 10—modulates receptor-mediated effects.

Chemical Groups and Abbreviations	Study Findings from Lines of Evidence for KC 10—Alters Cell Proliferation, Cell Death or Nutrient Supply		
	Epidemiological	Animal Biosassay	In Vitro
PFOA		Association: [61]	Association: [202]; [204]; [205]; [207]; [209]
Long-chain PFAS ^a^			
PFOS			Association: [203]; [207]; [208]
PFHxS		Association: [62]	
PFNA		Association: [61]	
8:2 FTOH			Association: [210]
Short-chain PFAS ^b^			
PFBS		Association: [62]	
PFHxA		Association: [61]	
6:2 FTOH			Association: [210]

^a^ The following long-chain PFAS are not presented in the table since no data were avaible to assess this key characteristic: PFDA, PFUnA, PFDoA, PFTrDA, PFTeDA, PFOSA, 8:2 monoPAP, 8:2 diPAP, 8:2 triPAP, 10:2 diPAP. ^b^ The following short-chain PFAS are not presented in the table since no data were avaible to assess this key characteristic: PFBA, PFPeA, PFHpA, GenX (HFPO-DA); PMOH, PMPP/ADONA, 4:2 FTOH, 4:2 diPAP, 6:2 diPAP.

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
