# Peer review of "Application of the Key Characteristics of Carcinogens to Per and Polyfluoroalkyl Substances"

_ijerph, 2020, doi:10.3390/ijerph17051668_

Round 1

Reviewer 1 Report

This is a compilation of data from 220 sources as tables and some conclusions. So it is not a regular review, but may be useful as a source for risk assessors. It is up to the policy of the journal, if this approach is in line with its principles.

This is basically a hazard based compilation not evaluating risks. Still it is somewhat annoying that doses (in vivo studies) or concentrations (in vitro studies) are not given any role at all (one exception, line 323 micromolar concentrations, which is high for PFAS). This is all the more important when there is evidence that these compounds are not electrophilic, mutagenic and genotoxic. To some extent this is compensated by comparisons with epidemiological studies, although also in these the reader is not given much help in assessing the levels of exposure. Also confounding is not touched at all (e.g. to what extent consumption of junk food is associated with both PFAS and obesity, the latter is clearly associated with several cancers and T2 diabetes). Mere presence of a chemical is mostly a measure of analytical chemist’s capabilities. Toxicity is not black-or-white, but highly dependent on doses and concentrations. Especially in in vitro studies it is possible to use concentrations that are not realistic in real life, and may cause many indirect effects (e.g. oxidative stress, inflammation). These issues on purely qualitative data should be made clear at least in the abstract and conclusions, if it is not possible to add dose considerations to the tables and/or the main text. Minor comment: check consistency, lines 356 and 362 Type 2 or type II? Hypersensitivity is Type I etc. (416, 424)

That said, the compilation is probably useful as a starting point. It reveals data gaps in existing literature. Because of the broad concept it is not possible to evaluate in detail, how reliable the interpretations of the authors are, but the text describing the overwhelming data is readable and I have no other major comments than the missing dose considerations.

Reviewer 2 Report

The manuscript provides a thorough review and assessment of the literature regarding the links between PFAS and the list of Key Characteristics (KCs) of carcinogens recently proposed as a framework for organizing and synthesizing data for cancer hazard identification. PFAS are a family of environmental chemicals for which there is significant concern in connection with their possible health effects and such an assessment is highly topical.

The authors have followed the methodology for the application of the concept of KCs of carcinogens as recommended in the relevant foundation publications. The manuscript initially introduces the environmental and public health relevance of PFAS as well as the concept of the KCs, and reviews the epidemiological and experimental animal-derived evidence on the carcinogenicity of this family of chemicals. Finally it goes on to systematically review the literature in relation to the ability of PFAS to induce each KC in turn, in each case ending with a conclusion regarding the adequacy of existing evidence based on pre-stated criteria.

Overall the manuscript constitutes a valuable piece of work that well deserves to be published. I have a small number of criticisms/suggestions which may help improve its value:

A Table summarizing the evidence regarding the various KCs (at least those links for which evidence is judged as at least suggestive) would be useful. Additionally a diagram indicating possible mechanistic links between such KCs would help to synthesise on a mechanistic basis the evidence and to better assess its strength in relation to the possible carcinogenicity of the chemicals in question. Early on (l. 59-62) reference is made to potential exploitation of the KCs concept with groups of PFAS, i.e. environmental mixtures. While, based on the literature cited, it appears that studies with mixtures of PFAS are lacking, no effort is made by the authors to approach this issue using the evidence coming from individual chemicals. I find the conclusion of paragraph 5.1 (l. 183-185), in combination with Table 2, confusing: While FTOHs/diPAPs are said to be metabolically converted to the corresponding acids and sulphonates, which are not electrophilic or metabolically activated, in the Table the former chemicals are indicated as having an “Association” with this KC (presumably implying that they undergo metabolic activation). Minor points:

- In Suppl. Table 1 the KCs are referred to by number (1-10). The corresponding numbers should be included in Table 1 of the main m/s.

- While the quality of the English language of the m/s is generally excellent, a thorough check for minor errors (mainly syntactic) would help to improve it further; e.g.

l. 3/4 – “use” of PFAS in discharges?; l. 165 – presumably if ONE (not two) type of evidence …;

l 201-203 – not “their carcinogenic potency” but “any carcinogenic hazard”;

l. 223-224, not “DNA methylation … in cord serum”; here and elsewhere in the section on DNA methylation the tissue of origin of the DNA should be clarified;

l. 342-344, “epidemiologic evidence … have been investigated”?

l. 665-667, I think that a different ref. is needed for Halifax Project Task Force publication (ref. 219?)
